# A Scoping Review on Lipocalin-2 and Its Role in Non-Alcoholic Steatohepatitis and Hepatocellular Carcinoma

**DOI:** 10.3390/ijms22062865

**Published:** 2021-03-11

**Authors:** Marinela Krizanac, Paola Berenice Mass Sanchez, Ralf Weiskirchen, Anastasia Asimakopoulos

**Affiliations:** Institute of Molecular Pathobiochemistry, Experimental Gene Therapy and Clinical Chemistry (IFMPEGKC), RWTH University Hospital Aachen, 52074 Aachen, Germany; mkrizanac@ukaachen.de (M.K.); pmasssanchez@ukaachen.de (P.B.M.S.)

**Keywords:** lipocalin-2, hepatocellular carcinoma, fatty liver disease, NASH, cancer, biomarker

## Abstract

Excess calorie intake and a sedentary lifestyle have made non-alcoholic fatty liver disease (NAFLD) one of the fastest growing forms of liver disease of the modern world. It is characterized by abnormal accumulation of fat in the liver and can range from simple steatosis and non-alcoholic steatohepatitis (NASH) to cirrhosis as well as development of hepatocellular carcinoma (HCC). Biopsy is the golden standard for the diagnosis and differentiation of all NAFLD stages, but its invasiveness poses a risk for patients, which is why new, non-invasive ways of diagnostics ought to be discovered. Lipocalin-2 (LCN2), which is a part of the lipocalin transport protein family, is a protein formally known for its role in iron transport and in inflammatory response. However, in recent years, its implication in the pathogenesis of NAFLD has become apparent. LCN2 shows significant upregulation in several benign and malignant liver diseases, making it a good candidate for the NAFLD biomarker or even a therapeutic target. What makes LCN2 more interesting to study is the fact that it is overexpressed in HCC development induced by chronic NASH, which is one of the primary causes of cancer-related deaths. However, to this day, neither its role as a biomarker for NAFLD nor the molecular mechanisms of its implication in NAFLD pathogenesis have been completely elucidated. This review aims to gather and closely dissect the current knowledge about, sometimes conflicting, evidence on LCN2 as a biomarker for NAFLD, its involvement in NAFLD, and NAFLD-HCC related pathogenesis, while comparing it to the findings in similar pathologies.

## 1. Introduction

LCN2, which is a protein encoded by the *Lcn2* gene on the human chromosome locus 9q34.11 and mouse chromosome 2, was first discovered as a protein associated with neutrophil gelatinase. Hence, its other name is neutrophil gelatinase-associated lipocalin (NGAL). It was purified by Kjeldsen and colleagues who established that its sequence did not match any known human protein but showed a high degree of similarity with the murine 24p3 protein, later proven to be its homologue [1]. Apart from the above names, it is also often referred to as 24p3, oncogenic lipocalin, siderocalin, 25 kDa α2-microglobulin-related protein, and uterocalin, indicating its omnipresence and pleiotropic functions in various tissues. Throughout this review, we will refer to this protein with the term LCN2.

LCN2 is a part of the lipocalin family of proteins with limited mutual sequence homology but a preserved tertiary barrel structure, consisting of eight-stranded anti-parallel β-barrel enclosed by two α-helices forming a hydrophobic pocket. Their primary role is the transport of small hydrophobic molecules including retinoids, steroids, and drugs [2]. The first role assigned to LCN2 was part of the innate immune response to bacterial infection where LCN2 serves as an iron-containing siderophores sequester [3]. It binds bacterial siderophores, thus, preventing the bacterial strategy of iron uptake, leaving them in need of nutrients and, in that way, reducing the duration of infection. Since it may be presumed based on the variety of names, its roles do not end with iron transport. Quite the opposite, its roles range from implication in cell apoptosis [4], lipid metabolism [5], cell migration [6], tumor invasiveness, and even metastasis [7]. It is even included in the regulation of receptors trafficking. For example, it is responsible for the epidermal growth factor receptor (EGFR) intracellular trafficking mediated by TGF-α [8]. It acts by binding to the intracellular domain of EGFR in late endosomal compartments, thus, inhibiting its lysosomal degradation. Constitutive activation of EGFR is widely known as an instigator of tumor cells proliferation, growth, survival, and migration, which is why it is speculated that the inhibition of LCN2 in this context could serve as a therapeutical target.

LCN2 can exist as either a 25-kDa monomer or a 56-kDa homodimer, which is the most abundant form in healthy individuals, or as a 130-kDa LCN2-matrix metalloproteinase 9 (MMP-9) complex. It has been proposed that dimers are the result of prolonged LCN2 storage in neutrophils that allow dimers to be formed before excretion [9]. This explanation comes from the fact that epithelia-derived LCN2 is primarily composed of monomers since it is being excreted directly, without storage [9,10]. However, to date, no differences in the pathophysiological functions between the monomer and dimer, except for faster clearance of the monomer from circulation, have been reported [11]. Regarding the LCN2/MMP-9 complex, research suggests that LCN2 can act as a supporting factor for MMP-9, making it less prone to hydrolysis and prolonging its activity [12]. The complex is present only in humans since a mouse homologue lacks Cys-87 residue, which forms a di-sulphide bond with an unidentified cysteine residue in pro-MMP-9. For the same reason, murine models also lack the homodimer form [13]. This fact, which is sometimes neglected, is of the most utter importance since the majority of the LCN2 research is done on murine models and could also account for the discrepancies in the results between the human and murine models presented below. There has been a great review written on the problematics by Bauvois and Susin [14]. One other form has been described lately, known as a 30-kDa isoform, which likely originates from different site-glycosylation [15]. Data are scarce regarding the N-glycosylated form of the LCN2 and, for now, it is only reported in literature that the glycosylation makes the protein more soluble, while it is not required neither for secretion nor exosome targeting [15].

## 2. Regulation of LCN2

The promoter for LCN2 contains binding sites for nuclear factor-κB (NF-κB) and CCAAT/enhancer-binding protein (C/EBP), which goes in agreement with its expression mostly being regulated by inflammatory pathways [16,17]. It has been reported that the mitogen-activated protein kinase (MAPK) pathway may cooperate with NF-κB to upregulate the expression of LCN2 [14]. The most profound effect on the expression LCN2 have proven to have the pro-inflammatory cytokines such as tumor necrosis factor-α (TNF-α), Interleukin 1β (IL-1β), and Interleukin 6 (IL-6) [2]. The LCN2 gene also contains cis-binding elements for transcription factors such as specificity protein 1 (SP1), polyoma enhancer activator (PEA3), lymphocyte function-associated antigen 1 (LFA-A1), and glucocorticoid receptors (GRs) [11]. Considering that these elements are tissue-specific, their presence allows for LCN2’s varying roles in physiological processes. However, new binding sites and also trans-acting elements have been found [18].

To this day, three LCN2 receptors have been discovered and characterized. One is the endocytic low-density lipoprotein receptor megalin, which binds the iron transporting LCN2 with high affinity and mediates its cellular uptake [19]. Even though megalin is a multi-ligand receptor, its higher-binding affinity toward LCN2 compared to other adipokines has been proven, but not completely explained. For now, it is considered that it results from the net positive charge displayed by LCN2, making it easy to differentiate among other adipokines. Another receptor is the solute carrier family 22, member 17 (SLC22A17), which is also known as the organic cation transporter (BOCT), 24p3R or neutrophil gelatinase-associated lipocalin receptor (NGALR) [20]. While megalin does not distinguish between apo and holo-forms of LCN2, SLC22A17 seems to be able to discriminate between the two and selectively mediate apoptosis and iron uptake in cancer cells [21,22]. A third receptor identified is the melanocortin-4 receptor that is particularly involved in the control of energy metabolism and expenditure [23]. Given that LCN2 is present in a broader range of tissues and possesses many more functions than can be explained by its binding to only these three receptors, it would be no wonder if some new potential receptors for LCN2 arise in the following years. For the moment, evidence points out that LCN2′s roles are a consequence of its lipophilic ligand transport [24]. For instance, lipocalins are known to transport fatty acids and retinoids, which, once inside the cell, can cause activation of signaling pathways or gene transcription [25]. However, it cannot be excluded that LCN2 possess the ability to induce a direct signal through binding to its specific, still undiscovered, membrane receptor.

## 3. The Expression of LCN2 in Tissues

LCN2′s expression starts in the foetal stage but is fairly low in an adult, healthy human. According to Human Protein Atlas, LCN2 protein is physiologically expressed in nine tissues of the adult body including kidney, cervix, bone marrow, nasopharynges, bronchus, stomach, spleen salivary gland, and skin. However, its RNA expression exceeds that number. It is typically excreted by neutrophils, basophils, ductal cells, pancreatic endocrine cells, exocrine glandular cells, and urothelial cells [Human Protein Atlas available from http://www.proteinatlas.org; last accessed 3 January 2021]. Importantly, all these cells are either specialized in an immune defence or serve as a first barrier (mucosal and physical) for pathogens confirming its role as a “primary defendant”.

Nonetheless, during pathological conditions, the number of cells secreting LCN2 grows. Due to the multitude of its roles, LCN2 shows significant upregulation in various benign and malignant diseases, non-even related to immunological defence. It has been shown that the upregulation of LCN2 is a common factor upon liver injury caused by factors such as excess alcohol consumption, and hepatectomy following hepatitis B and C virus infection [26]. Knowledge gained about LCN2 in the last decade detects its involvement in non-alcoholic fatty liver disease (NAFLD). This is a condition where hepatocytes serve as a major source of LCN2 [27]. However, the role of LCN2 in the pathology has not yet been fully elucidated, which is why, for this review, we aimed to collect current data on its expression in hepatic malfunction with emphasis on elucidating its dysregulation in non-alcoholic liver pathology and subsequent culmination in hepatocellular carcinoma (HCC).

## 4. Non-Alcoholic Fatty Liver Disease

NAFLD, defined as an extensive accumulation of fat in the liver, is a fast-growing form of new world’s disease caused by excess calorie intake and a sedentary lifestyle. Since NAFLD seems to be associated with classical indicators of metabolic syndrome (e.g., insulin resistance, Type 2 diabetes, and dyslipidemia), it is no wonder that LCN2 overexpression was lately defined as a hepatic manifestation of a metabolic syndrome [28].

On the molecular level, NAFLD is characterized by abnormal accumulation of triglycerides in hepatocytes. In a healthy homeostatic state, triglycerides are subjected to lipolysis in adipose tissue. Thus, their accumulation is being prevented. In the state of obesity, abundance of triglycerides and free fatty acids (FFA) causes impaired insulin signaling, resulting in abruption of lipolysis as well as excess delivery of FFA to the liver [29,30]. Another way of triglycerides accumulation, equally represented in the NAFLD pathogenesis, is de novo lipogenesis caused by excess glucose and fructose availability in the obese [31]. De novo lipogenesis is increased five-fold in non-alcoholic steatohepatitis (NASH) patients compared to healthy controls, independently by obesity [32]. Both mechanisms, though different, cause the same end result in the liver–lipotoxicity. Lipotoxic effect of the accumulated fatty acids can be observed in the metabolic reprogramming of mechanisms such as oxidative stress, autophagy, lipo-apoptosis, and inflammation [33]. FFA mediate their effects by dysregulating transcription factors, nuclear receptors, and membrane transporters of liver resident cells, causing cytokines mediated immune cell infiltration.

In NAFLD, during the stage of simple steatosis, the condition can be completely reversed through a simple change in lifestyle, including diet and physical exercise [31]. However, specific molecular changes and diverse immune cell activation are possible. These changes result in a progressive cascade of events causing diseases ranging from NASH to fibrosis, cirrhosis, and HCC (Figure 1). To make it clear, there are some guidelines on how to distinguish one phase from the other. For instance, NAFL, followed by lipid accumulation, caused inflammation and progresses to NASH [34]. While both NAFL and NASH show more than 5% hepatic steatosis, these two states can be easily distinguished histopathologically through the presence of lobular inflammation and hepatocyte ballooning with or without fibrosis in the latter. Lobular inflammation is categorized by infiltration of polymorphonuclear leukocytes, lymphocytes, and other mononuclear cells, which rarely include plasma cells and eosinophils [35].

A subsequent step through which severity of NAFLD can be assessed is fibrosis. It can even be said that this condition is the most important feature of NAFLD since it has been reported that it is the only independent predictor of liver-related mortality among all NAFLD pathological features [36]. Fibrosis can be described as a partial loss of liver function due to trans-differentiation of quiescent hepatic stellate cells (HSC) into fibrogenic myofibroblasts (MFBs) triggered by profibrogenic and pro-mitogenic mediators released from injured liver cells (e.g., hepatocyte and Kupffer cells) [37]. The pathogenesis includes extracellular matrix (ECM) remodeling mediated by release of collagens and glycosaminoglycans. Most potent transforming factors are proven to be the transforming growth factor-β (TGF-β) and platelet-derived growth factor (PDGF) [38]. The end stage of this process is classified as cirrhosis and characterized by the conversion of normal liver tissue to structurally abnormal architecture with fibrotic scarring and hepatocytes nodules [39]. With each step of the cascade above, liver function becomes impaired and it acts as a base for further pathologies, such as HCC.

This setting of dying hepatocytes, oxidative stress, DNA damage, and immense immune cell activation seems to act as a perfect foundation for liver carcinogenesis. In fact, almost half of cirrhotic patients develop HCC, but the mechanism of carcinogenesis is still not fully elucidated. Multiple reviews have been written on this subject, and all of them conclude that there ought to be an immense amount of research to clarify each step in the carcinogenesis [40,41,42].

## 5. LCN2 in NAFLD Pathophysiology

As previously mentioned, biopsies and histopathological studies on liver tissue need to be performed to distinguish between different NAFLD stages. In particular, this is required in late NAFLD diagnosis since most of the NAFLD patients fail to show any symptoms and are even often left undiagnosed [43]. There are a few non-invasive methods of NAFLD detection. For example, laboratory tests, imaging (ultrasonography, CT, MRI), and assessment of blood or urine biomarkers are diagnostic alternatives. However, they have all failed to cover and differentiate the whole spectrum of NAFLD, which is why biopsies still represent the golden standard [44]. That is why new non-invasive ways of determining the grade of the liver damage are urgently needed.

In this context, LCN2 has proven to be a good target to study as a possible biomarker for NAFLD and subsequent liver pathology (Figure 2). It has been proven by independent research teams that its high levels can indicate liver damage [45,46]. Since it can be detected in bodily fluids such as blood and urine, it can be routinely used as a part of laboratory tests.

To prove it, one study examined the levels of circulating LCN2 as well as its gene expression in obese women with NAFLD (with either NASH or simple steatosis) and normal liver [47]. The research concluded that both gene expression and protein levels were upregulated in obese women with NAFLD. However, gene expression correlated with simple steatosis while protein levels correlated with NASH. The same study showed that treatment with proinflammatory TNF-α, IL6, and resistin causes the upregulation of LCN2 in HepG2 cells [47]. Therefore, LCN2 is considered to be a liver’s protective response to inflammation. Another study, done on Chinese subjects, proved that LCN2 serum levels are elevated in patients with NAFLD as opposed to the control and that they highly correlate with both inflammation (C-reactive protein) and insulin resistance [48]. The study of Milner and colleagues showed that LCN2 levels correlated with the degree of liver inflammation and the stage of hepatic fibrosis as well as insulin resistance [49]. The diagnostic value of serum LCN2 was estimated by Xu and colleagues. Around 500 patients with either NAFLD (steatosis: *n* = 83, NASH: *n* = 277) or alcoholic fatty liver disease, together with a healthy control, were recruited for the study. The study showed that three variables, including serum LCN2 level, body mass index (BMI), and low-density lipoprotein (LDL) cholesterol, are positively correlated with NAFLD. The predicted value of LCN2 was calculated by multiple regression analysis, and the predicted value was applied to calculate the cut-off value using receiver operating characteristic (ROC) analysis. The area under the ROC curve of serum LCN2 was 0.987 with a specificity of 100% and a sensitivity of 93.5% for NASH diagnosis, and 0.977 with almost the same specificity and sensitivity for steatosis. Both cases showed a low rate of a false positive and a false negative. These data confirmed high diagnostic value of serum LCN2 in NASH and steatosis. However, although the serum LCN2 levels were compared between NAFL and NASH groups, ROC curve analysis was unable to establish an optimal cut-off value of serum LCN2 levels for distinguishing NASH from NAFL subgroups [50]. While all these studies portray LCN2 as a potential biomarker, they have all failed to prove that LCN2 can serve as a tool to differentiate the particular stages of NAFLD. Nonetheless, it would be of high importance to pinpoint its role in the pathogenesis. While, as mentioned above, human studies focus on LCN2 as a biomarker, there had been excessive studies to address LCN2’s role in NAFLD pathogenesis performed both in in vitro and in vivo models.

For instance, classical approach of LCN2 overexpression in hepatocytes done by Xu and colleagues resulted in lipolysis and fatty acid oxidation. Through this study, it was demonstrated that LCN2 overexpression prevents de novo lipogenesis, lipid peroxidation, and apoptosis, thus, preventing steatohepatitis [51]. This is one of the many studies that imply that LCN2 regulates liver lipid homeostasis.

A study done by Semba and colleagues aimed to detect if LCN2 can be a factor in the differentiation of simple steatosis and NASH [52]. For that purpose, DNA microarray analysis of the liver transcriptomes, RT-qPCR, and immunohistochemistry were done on murine models of simple steatosis (dd Shionogi mice) and NASH (fatty liver Shionogi mice). This study showed that LCN2 is overexpressed in mice with NASH, together with chemokines CXCL1 and CXCL9, while their overexpression was missed in mice with steatosis. What seems to be most interesting, all three proteins have specific localization, likely correlating with their role. LCN2 seems to be localized in hepatocytes and correlates to inflammatory cell clusters due to its implication in neutrophil signaling.

Another study from our group showed a hepatoprotective role of LCN2 both in vivo and in vitro. We fed wild type (WT) and *Lcn2*-deficient (*Lcn2*^−/−^) mice with a methionine and choline deficient (MCD) diet as a nutritional model of NASH. We found that LCN2 maintains lipid homeostasis through, among others, the induction of proteins important for lipid droplet formation named Perilipin 5 (PLIN5). It became clear that depletion of LCN2 or PLIN5 prevented normal intracellular lipid droplet formation in murine models as well as cell lines. The homeostasis was restored after transfection or adenoviral vector infection, which only confirmed the importance of LCN2 in liver protection [53]. In another previous investigation, we also performed a comparative analysis of *Lcn2^−/−^* and WT, high fat diet-fed mice. We managed to detect the proteins BRIT1/MCPH1, HMGB1, FABP5, and PLIN5 as important factors in LCN2-mediated lipid homeostasis. Besides that, it was shown that LCN2 increased mitochondrial activity, intra-mitochondrial chelatable iron pool, and a peroxisome number, suggesting that it is possible for LCN2 to act as a sensor that measures fat content and adjusts homeostasis by modifying peroxisome numbers and/or mitochondrial activity [54].

Another role has been assigned to LCN2 by Ye and colleagues. In their study, they induced NASH by either a high fat, high cholesterol (HFHC) diet or an MCD diet in mice. They could show how LCN2 mediates NASH by promoting neutrophil-macrophage crosstalk via the induction of CXCR2 [55]. It was noticed that the infiltration of neutrophils and macrophages was substantially attenuated by genetic depletion of *Lcn2*, but was augmented by chronic infusion of recombinant LCN2, thus, promoting inflammation. Mice lacking CXCR2 are resistant to LCN2-evoked liver inflammation. Even though the mechanism by which LCN2 may upregulate CXCR2 is still not known, the authors believe that it could be through an NF-κB-dependent mechanism.

As mentioned above, major sources of NAFLD pathogenesis seem to be glucose and fructose lipogenesis-derived FFA. However, a series of recent studies have indicated that sugars can induce NAFLD by means independent of de novo lipogenesis. A study conducted in our group showed how excess fructose leads to hepatic steatosis. However, in this context, fructose appears to directly affect liver homeostasis, thereby, manipulating fat metabolism [56]. Fructose might disturb liver homeostasis by promoting lipid uptake into the liver, while LCN2 counteracts lipid uptake. The same study showed there are potential differences between the sexes in LCN2-mediated lipid metabolism. This finding is of the utmost importance because it also shows a potential influence of oestrogens on LCN2-mediated lipid homeostasis. This goes in agreement with another study made by Alwash and colleagues who fed rats a fructose high diet that provoked a gradual increase of the LCN2 level over the course of eight weeks [57]. LCN2 levels seemed to correlate both with increased indicators of oxidative stress and mitochondrial dysfunction. Their model suggests a hepatoprotective role of LCN2 since its expression in a rat model seems to be provoked by endotoxins and inflammatory cytokines, as mentioned by many more [46,58]. Although these studies have managed to portray LCN2 as a hepatoprotective element in the liver, they have not managed to provide an exact mechanism, or mechanisms of its action regarding NAFLD pathology. As a matter of fact, they have not even explained whether LCN2 is a mere consequence of the pathology or its driver.

## 6. The Presence of LCN2 in Cancer

Cancer is a leading cause of mortality worldwide accounting for the efforts to elucidate the implication of LCN2 in the given pathology [World Health Organization available from [https://www.who.int/, last accessed 14 February 2012]. LCN2 is no stranger to multiple cancer pathologies. There is ample evidence for its implications in breast [59], thyroid [60], colon [61], endometrial [62], ovarian [63], lung [64], and liver cancer, as well as various adenocarcinomas [65] and leukaemia [66]. It shows significant upregulation in most of the above, but what seems to be interesting is that a decrease in expression is present in some metastatic tumors compared to primary tumors [67]. In this section, we focus on the implication of LCN2 in tumorigenesis.

Regarding its role in particular types of cancer, research seems to focus on its implication in epithelial-to-mesenchymal transition (EMT) and cancer cell proliferation. LCN2 seems to mediate tumorigenesis by facilitating tumor invasiveness. For instance, a breast cancer study shows that overexpression of LCN2 upregulates mesenchymal markers like vimentin and downregulates epidermal ones, such as E-cadherin [59]. In other words, LCN2 causes changes, which are the hallmark of EMT. In line with this assumption, LCN2 silencing inhibits breast cancer cell migration and mesenchymal phenotype [59]. Similar research done on oesophageal cancer shows how LCN2 mediates the activation of the mitogen-activated protein kinase (MEK/ERK) pathway and leads to an increase of MMP-9 activity [66]. These findings suggest LCN2 as a strong tumorigenesis promoter.

While LCN2 has proven to promote tumorigenesis in the previously mentioned cancers [59,60,61,62,63,64,65,66,67] as well as in esophageal squamous cell carcinoma [68], it seems to serve as a metastasis suppressor in other types of cancer. It was reported that overexpression of LCN2 in the pancreas correlates with high expression of E-cadherin and it disables EMT. To align with the previous findings, LCN2 is significantly downregulated in primary malignant and metastatic tissues of oral cancer in comparison to normal tissues. In this case, the mechanistic target of the Rapamycin (mTOR) pathway is activated when LCN2 is silenced, leading to oral cancer progression [69].

Regarding LCN2, there are sometimes conflicting results within one type of tumor, as it will be further analyzed in the section of liver cancer. As it can be seen, findings on LCN2 in cancer portray its paradoxical effects on individual cancer types that need to be further investigated before any kind of therapy targeting LCN2 is planned.

These examples are merely the tip of the iceberg in conflicting evidence regarding LCN2 in cancer. An extended overview of the implication of LCN2 in various other cancer types is presented in Table 1.

It is still unknown what roles the LCN2 exerts in the tumor microenvironment, but new studies suggest that its expression strongly correlates with immune cell infiltration; mainly with the infiltration of neutrophils and type T17 helper cell, which was detected across 32 cancer types [79]. In this study, a Gene Set Enrichment Analysis (GSEA) showed that the expression of LCN2 was related to retinol metabolism, drug metabolism cytochrome P450, and metabolism of xenobiotics by cytochrome P450. One more emerging role of LCN2 in cancer pathogenesis is related to its first known function, which is called iron transport. There had been an abundance of research done on the tumor microenvironment and attempts to discover in which way the tumor steers the immune cells around it [80,81]. Since tumor is in high demand of iron for its fast expansion, it would be easy to speculate that it has a mechanism, which enables it to sequester iron from the environment. Jung and colleagues tried to explain and relate the underlying mechanism to LCN2 [82]. They described how tumor-associated macrophages (TAMs), known as a polarized macrophage phenotype that facilitates tumor growth, handle iron transport when “hijacked” by the tumor. The main player is a lipid mediator sphingosine-1-phosphate (S1P), released by apoptotic tumoral cells, which causes LCN2 transcription in macrophages. They also reported that the overexpression of LCN2 in tumors could serve as a mechanism to facilitate the delivery of iron into the tumors via 24p3R to meet their iron demand. LCN2-deficient tumors display substantially reduced iron content, partially confirming this report [83].

## 7. LCN2 and Its Significance in Hepatocellular Carcinoma

Regarding the liver, it has been known for some time now that, except for alcohol consumption and hepatitis virus-caused liver cancer, obese patients and patients with NAFLD possess greatly increased risk of HCC. To put into perspective LCN2’s implication in the pathogenesis of NAFLD-induced HCC, it is important to take into account all aspects of its primary cause, namely obesity.

On an organism scale, obesity is found to promote tumorigenesis through elevated levels of adipokines, growth factors, steroid hormones, chronic inflammation producing reactive oxygen species (ROS), and a change in gut microbiota [84]. For example, it has been proven that accumulated visceral fat, which is a hallmark of obesity, correlates with primary tumor recurrence, poor prognosis, and chemotherapeutic resistance [85,86] In that setting, adipose tissue plays a significant role by excreting adipokines, with LCN2 being one of them. Some authors even suggest that LCN2 expressed by adipose tissue causes the activation of both MMP-2 and MMP-9, which are promoters of tumorigenesis, and they found especially increased circulating concentrations of the LCN2/MMP-9 complex, which is known for its tissue remodeling properties [87]. These are some of the examples that portray how obesity-caused LCN2 is a part of the tumorigenesis process. It is important to mention a study on adipose-derived LCN2 done on murine models that brought evidence on both sex-specific and tissue-specific roles of LCN2 [18]. It was concluded that adipose excreted LCN2 contributes to dysregulation of metabolism in females, and liver LCN2 seems to act on metabolic traits of males, specifically on insulin resistance. Adipose LCN2 also negatively regulates its receptor low density lipoprotein receptor-related protein 2 (LRP2) and repressor ERα in a female-specific manner [18]. Even though the study did not analyze sex specificity of LCN2 in carcinogenesis, it has pinpointed an important aspect of LCN2′s role, which will be important to closely look into the future since it might highly influence this process.

Clinical studies have shown that the risk of HCC rises drastically in NAFLD patients with underlying cirrhosis [88]. A systematic review done by Freeman shows that the epidemiologic evidence gathered between 1992 and 2011 supports an association between NAFLD or NASH and an increased HCC risk that seems to be predominantly limited to individuals with cirrhosis. Statistically, approximately half of HCC cases occur in patients who have a profile suggestive of fatty liver disease [89].

Regarding the role of LCN2 in non-alcoholic HCC, there has not been in-depth research conducted to elucidate its implication in the pathogenesis (Table 2). This part is dedicated to closely dissect current findings in the field.

To begin with, most recent findings portray LCN2 as a novel diagnostic tool for HCC [90]. This study, which was conducted on 300 subjects, reported that serum LCN2 levels showed significantly high diagnostic performance of HCC in comparison to commonly used α-fetoprotein (AFP). Moreover, they found that LCN2 levels are able to discriminate between HCC and liver cirrhosis subjects, and they also proved that LCN2 levels are not suitable to distinguish between early and late HCC. This goes along with an Egyptian study showing how even urine LCN2 can serve as a marker in HCC diagnosis [91]. In the respective study, LCN2 levels were significantly elevated in patients with HCC in contrast to the control, cirrhotic patients, and patients with chronic hepatitis. One different study, based on microarray analyses, focused on the characterization of genes highly expressed in HCC [92]. It was detected that LCN2, alongside several other genes that encode secreted and membrane-bound proteins (e.g., AFP, Glypican-3, GPC3, Immunoglobulin superfamily member 1, IGSF1, Prostate-derived Ste20-like kinase, PSK-1), is overexpressed at the HCC site. However, even though the underlying evidence marks these genes as potential diagnostic markers, there has been a varying difference of gene expression between the samples, which is a sign that the cancer origin plays a role in a later expression profile of HCC. Based on the fact that expression profiles between HCC samples differ based on the cancer origin, it would be interesting to see whether LCN2 is a marker, which rises during NAFLD-induced HCC pathogenesis, and, if so, solely or as a part of a general modified gene profile. In Figure 3, we collected the roles of LCN2 in the pathogenesis and diagnosis of HCC.

Not only can LCN2 potentially serve as diagnostic marker, but it can possibly serve as a predictor of survival. That conclusion was drawn from a study which aimed to detect if LCN2 serum levels show any correlation with HCC patient’s mortality [97]. As a matter of fact, it was presented that values of serum LCN2 concentrations above 217.5 μg/L at the time of diagnosis are associated with higher mortality. Even though the study has been done on a small sample size, it sheds a new light on the role of LCN2 in HCC. Some authors have also recognized the importance of LCN2-associated proteins. For example, research done by Zhang and colleagues has provided evidence of not only LCN2 overexpression in HCC but also the overexpression of its receptor, NGALR [95]. This upregulation correlated positively with the tumor stage, vascular invasion status, and tumor recurrence in patients with HCC.

While most of the studies focus on confirming LCN2 as a biomarker for HCC, there had been a few studies focusing on assigning it a role. One of those studies found LCN2 overexpression in murine HCC livers nodules, both on a transcriptional level and a protein level. High expression seems to correlate with known HCC marker AFP and is also co-localized with Myeloperoxidase (MPO), which is a neutrophil marker, proving that hepatocytes are not the only source of LCN2 in HCC. Another interesting finding was that LCN2-expressing cells in human HCC samples are not CD90 expressing cells, while there is a strong correlation between the two in a murine model. As the authors state, this co-localization is likely a proof of species-dependent mechanism of LCN2 in the process of cell differentiation during the HCC progress [53].

Levels of LCN2 in HCC have been found to positively correlate with 3,3′,5-triiodo-L-thyronine (T3) and its receptor (TR), which led to a study that detected a TR element binding site on LCN2′s promoter [6]. Except for that, the study showed how overexpression of LCN2 enhances tumor cell migration and invasion, and, conversely, its knockdown in human cell lines suppressed their migration and invasion, both in vitro and in vivo. The LCN2 overexpression experiment showed that it is a factor in EMT. As for the mechanism, it seems that the overexpression of LCN2 leads to suppression of E-cadherin and increase in p-Met and p-FAK protein levels, while its depletion rescues E-cadherin and suppressed p-Met and p-FAK protein levels as well as MMP-9 and MMP-2 activities. These findings depict LCN2 as a factor in tumor progression and invasiveness.

However, several other studies suggest that LCN2 has the opposite effect on liver cancer cells. LCN2 seems to have an anti-tumorigenic effect on liver cancer cells [4,7,95].

Regarding the signaling in which LCN2 seems to be included, the first finding indicated that LCN2 modulates EMT negatively, at least in part, through an EGF (or TGF-β1)/LCN2/Twist1) signaling pathway [7]. Since EMT is a critical point in tumor invasion, this finding presents LCN2 as a metastasis suppressor, and even makes it a potentially good therapeutic target. Previous research, done by the same laboratory, also classifies LCN2 as a tumor suppressor since is seems to diminish proliferation and invasion of HCC cells through the blockade of JNK and PI3K/AKT signaling. The groups of Lee and Chien reported that LCN2 causes the HCC cells apoptosis. An overexpression experiment in SK-Hep-1 and Huh-7 cells showed that LCN2 significantly decreases cell viability, arrests the cell cycle at the sub-G1 phase, causes DNA fragmentation, and activates caspase as well as apoptotic-related or anti-apoptotic-related proteins [4,95].

Even though research on LCN2 in HCC gives conflicting results, some treatments for HCC seem to include LCN2. The implication of LCN2 in the mechanism of Sorafenib susceptibility in liver cancer has been discovered a few years ago [96]. While exploring the mechanism in which Sulfatase 2 (SULF2) mediates Sorafenib susceptibility, it has been concluded that the mechanism involves deregulation of LCN2. This is one of the first studies which connects LCN2 to treatment sensitivity and marks it as a possible therapeutic target of liver cancer.

## 8. Conclusions

This review aimed to gather the up-to-date findings on LCN2 in NAFLD and HCC pathology. The role of this lipocalin as a biomarker seems very straightforward since independent research have implied LCN2 levels as a diagnostic tool to differentiate several stages of NAFLD as well as HCC from the previous tool. However, it is still too early to use LCN2 as a universal NAFLD biomarker. Moreover, whether urine or serum levels ought to be used and is there a protein whose co-expression with LCN2 might improve possibilities in differentiation of NAFLD stages still needs a further investigation.

One of the toughest challenges for all researchers in this area will be to comprehend a very un-predicted and sometimes even contradicting behavior of LCN2 in NAFLD-HCC pathology. When we try to understand the underlying cause of the conflicting roles of LCN2 in NAFLD and NASH-related HCC, it is hard not to speculate that this range of behaviors is the product of either its special form (monomer/dimer/complex) of LCN2, microenvironment (growth factors, cytokines, and immune cells), or even sex-specific characteristics (oestrogen) present in different NAFLD-HCC models used. Each of these items needs to be closely studied to get the wholesome picture of LCN2′s biological activities. In this context, it would be of great importance to develop more accurate NAFLD-HCC models better reflecting the human pathogenesis.

The main conclusion that can be drawn from the studies gathered in this review is that there is still a lot unknown about LCN2, not only in NAFLD and HCC, but also other pathologies. Whether the dysregulation of LCN2 levels reflect its instrumental role in the pathogenesis of the disease or act as a response to it, is yet to be elucidated. Information gathered in this review could provide a guideline for the direction in which the research of NAFLD-HCC pathophysiology should be directed to uncover the molecular mechanisms underlying LCN2 activity and making its use as a biomarker routinely available.

## Figures and Tables

**Figure 1 ijms-22-02865-f001:**
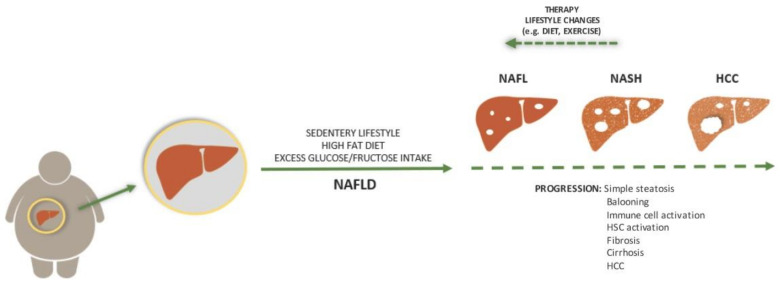
Aetiology and progression of non-alcoholic fatty liver disease (NAFLD). In most cases, NAFLD is a consequence of sedentary lifestyle, a high fat diet, and excess glucose/fructose intake. While the state can be completely reversed with a lifestyle change or therapy, its progression is driven by pathological processes that result in histologically visible steatosis, ballooning, fibrosis, and cirrhosis. Each step is accompanied by immune cell activation while later steps include hepatic stellate cell activation. The end stage of this process can be hepatocellular carcinoma (HCC).

**Figure 2 ijms-22-02865-f002:**
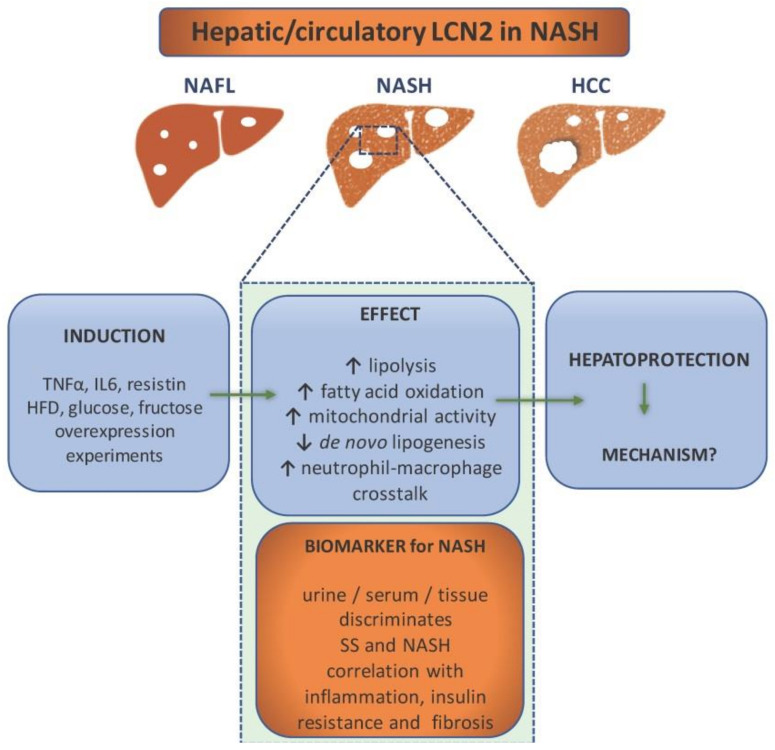
Lipocalin-2 (LCN2) as a component in non-alcoholic steatohepatitis (NASH) pathology and diagnostics. NASH patients show elevated levels of LCN2 in urine, serum, and liver tissue, which makes it a potential biomarker for NASH. It is able to discriminate between simple steatosis (SS) and NASH, while the level of LCN2 correlates with inflammation (e.g., increase of CRP), insulin resistance, and fibrosis. It is induced by proinflammatory cytokines, high fat diet, glucose, and fructose overconsumption. Overexpression experiments have implicated LCN2 in the enhancement of lipolysis, fatty acid oxidation, mitochondrial activity, and immune cells crosstalk. In addition, LCN2 upregulation is associated with reduced de novo lipogenesis. All evidence points out that its role is hepatoprotective. However, the exact mechanisms are ought to be found.

**Figure 3 ijms-22-02865-f003:**
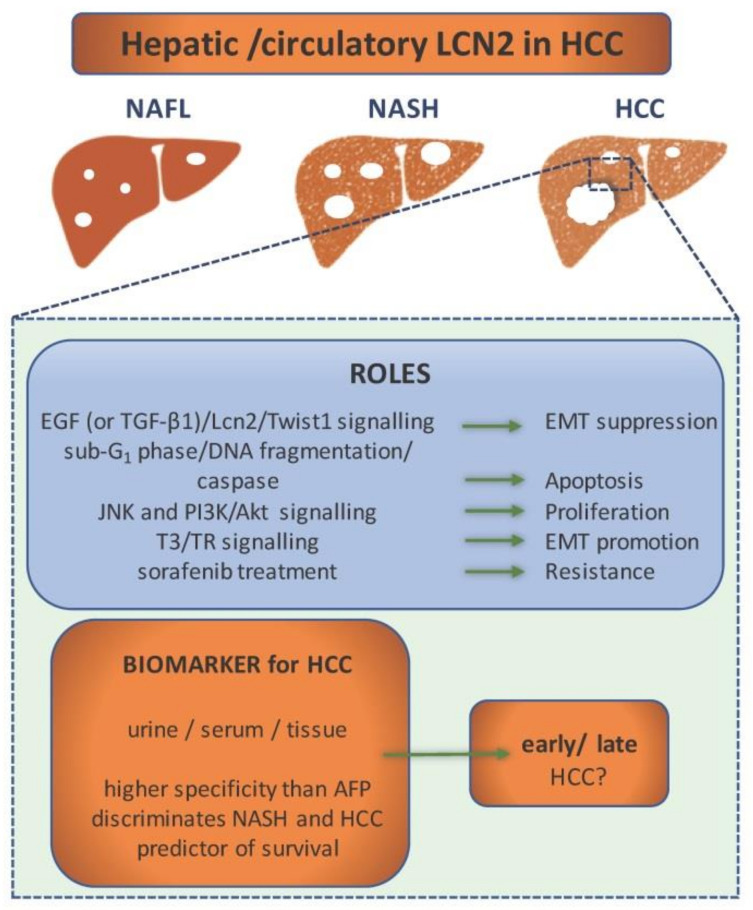
LCN2 as a component in hepatocellular carcinoma (HCC) pathogenesis and diagnostics. In the pathogenesis of HCC, LCN2 is included in various signalling pathways that regulate processes as epithelial-to-mesenchymal transition (EMT), apoptosis, proliferation, and chemoresistance. Depending on the activated signalling pathway, it can either act as a tumor suppressor or promoter. Altered expression level of LCN2 in urine, serum, and tissue during the pathogenesis of HCC makes it a potential biomarker. It shows higher specificity in diagnosis of HCC than α-fetoprotein (AFP). LCN2 levels discriminate HCC from other non-alcoholic fatty liver disease (NAFLD) stages and can even predict patient survival. However, it cannot discriminate between early and late HCC.

**Table 1 ijms-22-02865-t001:** Selected experimental and clinical findings associated with LCN2 in cancer models, tissues, and cell lines.

Type of Cancer	LCN2 Expression	Model	Major Findings	Function of LCN2	References
**Adeno-** **carcinoma**	upregulated	A549 cells and MCF7 cells treated with MK886	Apoptosis induced by treatment with MCF7 was accompanied by a dose- and time-dependent increase of *LCN2* mRNA levels	Data indicate that, although the induction of LCN2 correlates with apoptosis, induction represents a survival response	[65]
**Thyroid**	upregulated	siRNA knockdown in FRO cell line	*LCN2* knockdown blocks the ability of FRO cells to form colonies in soft agar and tumours in nude mice and induces apoptosis	LCN2 is a survival factor for thyroid neoplastic cells. Data suggests that NF-κB contributes to thyroid tumour cell survival by controlling iron uptake via LCN2	[60]
**Breast**	upregulated in tissue and urine	Breast cancer cell lines MCF-7 and MDA-MB-231 transfected with siRNA: Overexpression study on the same cell lines	Overexpression of LCN2 leads to an increase in mesenchymal factors (vimentin and fibronectin) and decrease in epithelial (E-cadherin). Silencing inhibits cell migration and reduces ER-α expression MCF-7 tumours revealed that the LCN2-overexpressing ones exhibited increased growth rates that were accompanied by increased levels of MMP-9, increased angiogenesis, and an increase in the tumour cell proliferative fraction	LCN2 promotes breast cancer progression LCN2-MMP-9 complex is facilitating angiogenesis and tumour growth	[59,70]
**Esophageal**	upregulated	EC109, SHEE, SHEEC, EC8712, KYSE150, KYSE180, and TE3 cell lines	LCN2 increases MMP-9 and phospho-ERM (phospho-ezrin/radixin/moesin), decreases phospho-cofilin and cytoskeleton F-actin rearrangement in oesophageal squamous cell carcinoma cells	LCN2 promotes the migration and invasion of oesophageal squamous cell carcinoma cells through the ERK1/2 pathway	[68]
**Ovary**	upregulated	HEY, PEO.36, SKOV3, OVCA433, and OVHS1 cell lines	Downregulation of *LCN2* expression correlates with the upregulation of vimentin expression, enhanced cell dispersion, and downregulation of E-cadherin expression	LCN2 is associated with an epidermal growth factor that induced EMT	[63]
**Endometrium**	high expression of LCN2 and vascular endothelial growth factor (VEGF), high LCN2 serum levels in cancer patients	HHUA and RL95-2, and LCN2-low-expressing cell line HEC1B	Effects of LCN2 silencing on cell migration, cell viability, and apoptosis under various stresses, including ultraviolet irradiation and cisplatin treatment	LCN2 was involved in the migration and survival of endometrial carcinoma cells under various stresses in an iron-dependent manner. The survival function of LCN2 may be exerted through the PI3K pathway and suppression of the p53-p21 pathway	[62]
**Colon**	upregulated	SW620-OB, SW620-LCN2 (5 × 106), SW480-SHB, and SW480-sh-LCN2 cells were inoculated subcutaneously into the BALB/c nude mice Knockdown of LCN2 using siRNA in colecteral cancer cells (CRC) cells LCN2 overexpression or antisense LCN2	LCN2 blocked cell proliferation, migration and invasion in vitro and in vivo, and inhibited translocation of NF-κB into the nucleus LCN2 negatively modulated proliferation, EMT, and energy metabolism in CRC cells Overexpression altered subcellular localization of E-cadherin and catenins, decreased E-cadherin-mediated cell-cell adhesion, enhanced cell-matrix attachment, and increased cell motility and in vitro invasion. Silencing aggregated a growth pattern and decreased in vitro invasion. These effects were mediated through the alteration of the subcellular localization of Rac1	LCN2 suppresses metastasis of colorectal cancer LCN2 negatively regulates cell proliferation and EMT through changing metabolic gene expression in colorectal cancer increasing proliferation and metastasis LCN2 decreases E-cadherin-mediated cell-cell adhesion and increases cell motility and invasion	[61,71]
**Lung**	upregulated	Downregulation by shRNA Knockdown by siRNA in lung cancer cell line A549	Depletion of *LCN2* expression decreased the ability of cell proliferation and induced cell apoptosis The radiosensitivity of these cells was enhanced	Downregulation of *LCN2* suppresses the growth of human lung adenocarcinoma through oxidative stress involving Nrf2/HO-1 signalling LCN2 increases lung cancer cells radio-resistance	[64,65]
**Chronic** **Myeloid** **Leukemia**	upregulated	*LCN2* mRNA in blood samples and protein in sera	A highly significant increase of mRNA expression and protein secretion was shown in patients at diagnosis	LCN2 play an important role in the physiopathology of CML	[66]
**Oral**	significantly downregulated in primary malignant and metastatic tissue	shRNA-mediated knockdown of *LCN2* was carried out in the SAS cell line	Knockdown increased oral cancer cell proliferation, survival, and migration. Silencing of *LCN2* activated mTOR signalling and reduced autophagy.	Downregulation of *LCN2* activates the mTOR pathway and helps in the progression of oral cancer. Silencing of *LCN2* increases oral cancer cell proliferation and survival Levels of LCN2 and the LCN2/MMP-9 complex may be useful in non-invasively monitoring OSCC progression and migration	[69,72]
**Kidney**	upregulated	CAKI 1, 786-O, A498, and RCC4 cell lines were subjected to treatment with iron free or loaded LCN2	Iron-free LCN2 reduced migration and matrix adhesion. In contrast, stimulation with iron loaded LCN2 enhanced migration and adhesion.	Iron load defines the pro-tumour characteristics of LCN2 in renal cancer	[73]
**Pancreatic**	high in serum (ELISA) and tumour tissue	LCN2 overexpression in pancreatic cell lines. Cells were subsequently injected into the subcapsular region of the nude mice pancreas. *LCN2* expression was downregulated by shRNA in pancreatic ductal adenocarcinoma cells (BxPC3 and HPAF-II); overexpression of LCN2 in the same cell lines	LCN2 overexpression (MIAPaCa-2 and PANC-1) significantly blocked cell adhesion and invasion in vitro, reduced Focal adhesion kinase (FAK) phosphorylation, potently decreased angiogenesis in vitro partly through reduced VEGF production Downregulation significantly reduced attachment, invasion, and tumour growth in vivo. The opposite results were found by LCN2 overexpression.	LCN2 acts as suppressor of invasion by suppressing FAK activation and inhibits angiogenesis partly by blocking VEGF LCN2 plays an important role in the malignant progression of pancreatic ductal carcinoma	[11]
**Gastric**	high in tumour tissue and serum	*LCN2* gene silencing in MGC-803 and SGC-7901 cells by *LCN2*-siRNA; cells were subsequently used for xenograft model in nude mice MGC-803 cells were treated with siRNA against LCN2 and also implanted into nude mice	The mice experiment showed that LCN2 gene silencing inhibited the proliferation and tumorigenicity of the MGC-803 and SGC-79 *LCN2*-siRNA cells exhibited inhibited proliferation, enhanced apoptosis, decreased expressions of NF-κB and Bcl- 2. Respective cells showed repressed tumorigenicity in vivo.	*LCN2* gene silencing inhibits proliferation and promotes apoptosis of human gastric cancer cells *LCN2* gene silencing inhibits proliferation and promotes apoptosis of MGC-803 cells	[74,75]
**Prostate**	high in tumour tissue and cell lines	*LCN2* knockdown in prostate cancer cells (PC3, DU145) by shRNA	Knockdown of *LCN2* suppresses growth and invasion of prostate cancer cells	LCN2 might play an important role in regulation of proliferation and invasion of human prostate cancer	[76]
**Cholangio-** **carcinoma** **(CCA)**	upregulated	Human RMCCA-1 cell line subjected to LCN2 downregulation by siRNA Human CCA cell lines were subjected to LCN2 knockdown and overexpression	*LCN2* knockdown suppressed invasion by reducing LCN2/MMP-9 complex formation *LCN2* knockdown inhibited CCA cell growth in vitro and in vivo through induction of the cell cycle arrest at G0/G1 phases and repression of EMT; overexpression of LCN2 in CCA cells increases cell metastatic potential	LCN2 promotes the invasiveness of the cholangiocarcinoma cells by forming a complex with MMP-9 LCN2 is a promising target for CCA treatment and bile LCN2 level is a potential diagnostic marker for CCA	[77,78]

**Table 2 ijms-22-02865-t002:** Selected experimental and clinical findings associated with LCN2 in hepatocellular carcinoma.

Species	Model/Sample	Experiment	Major Findings	Conclusions	Reference
**Human**	Serum from healthy individuals, patients with HCC or patients with cirrhosis	300 subjects were subjected to routine laboratory tests	LCN2 levels greater than 225 ng/mL have a higher diagnostic performance in HCC patients and are more accurate in differentiation between cirrhosis and HCC patients than α-fetoprotein (AFP)	LCN2 is a good candidate for HCC diagnosis and screening	[90]
**Human**	Tissue and serum samples from HCC patients and healthy individuals	Tissues were subjected to immunostaining and serum to Western blot analysis	Strongly elevated expression of LCN2 in diseased human liver instead of in a uniform pattern. All cells positive for either AFP or myeloperoxidase (MPO) were also strongly positive for LCN2.	LCN2 is pleiotropic, possibly participating in multiple functions in the tumor microenvironment, such as damage response, immunity, and differentiation	[53]
**Human**	HepG2, Huh7, SK-HEP1, and J7 HCC cell lines	HepG2 and J7 cell lines were stably transfected with stably transfected TRα1, Huh7, and J7 cell lines overexpressing LCN2	LCN2 is positively regulated by T3/TR. Overexpression of LCN2 enhanced tumor cell migration and invasion both in vitro and in vivo. LCN2-induced migration occurred by activating the Met/FAK cascade	T3/TR has a potential role of in cancer progression through regulation of LCN2 via the Met/FAK cascade	[6]
**Human**	THLE-2, HepG2, Hep3B, PLC/PRF/5 (Alexander cells), SH-JI, and SK-HEP-1 cell lines	Adenoviral transduction of *Lcn2* and knockdown of *Lcn2* by short hairpin RNA (shRNA)	Adenoviral upregulation of LCN2 causes the downregulation of epithelial-to-mesenchymal markers, while silencing reverses that effect	LCN2 negatively modulates the EMT in HCC through the epidermal growth factor *(TGF-β1)/Lcn2/Twist1* pathway	[7]
**Human**	Huh-7 and SK-Hep-1 cell lines	Cells were transfected with plasmids encoding full-length LCN2	LCN2 overexpression dramatically inhibited cell viability, induced apoptosis features reflected in cell-cycle arrest in sub-G_1_ phase, DNA fragmentation, and condensation of chromatin	LCN2 induces apoptosis in human hepatocellular carcinoma cells by activating mitochondrial pathways.	[4]
**Human**	Urine of HCC patients, patients with chronic viral hepatitis and cirrhotic patients	Urinary LCN2 levels were measured by an enzyme-linked immunosorbent assay	Urinary LCN2 content can discriminate between HCC and cirrhosis	Urinary LCN2 is a possible diagnostic marker for HCC patients	[91]
**Human**	102 primary HCC tissues, 74 nontumor liver tissues, seven benign liver tumor samples, 10 metastatic cancers, and 10 HCC cell lines	DNA microarray analysis done on tissues and cell lines	LCN2 is one of the top 10 genes overexpressed in HCC	This research is a step to define new candidate oncogenes and therapeutic targets in HCC	[92]
**Human**	25 hepatocellular carcinoma patients, hepatitis C patients, and 25 healthy subjects as a control	Measurements for hepatitis B surface antigen, hepatitis C antibodies, AFP, MMP-9, TIMP-1, and LCN2	Increased levels of LCN2 in HCC and HBV patients	LCN2 can be used as a future diagnostic marker with better sensitivity and specificity than MMP-9 for the progression of HCC	[93]
**Human**	Tumor tissues from 138 patients who underwent curative resection of HCC	Immunohistochemistry on tumor tissues	LCN2 and NGALR are both upregulated in HCC tissues and are associated with vascular invasion, tumor, nodes and metastasis (TNM) stage, tumor recurrence, and overall survival	LCN2 and NGALR expression might be served as novel prognostic factors and potential therapeutic targets in HCC	[94]
**Human**	HCC tissues and corresponding non-neoplastic liver tissues. Chang liver and SK-Hep1 human HCC cells	Tissue microarray experiment and analysis of *Lcn2* expressing HCC cells	Significant increase in LCN2 levels in human HCC tissues compared with non-tumor liver tissues. Ectopic expression of LCN2 in HCC cells significantly inhibited the growth of HCC cells in vitro and in vivo, reduced the invasive potential of cells, and inhibited the expression of MMP-2 partly through JNK PI3K/AKT signalling.	LCN2 inhibits the proliferation and invasion of HCC cells through a blockade of JNK and PI3K/AKT signalling	[95]
**Human**	55 cases of biopsied tissues for HCC, liver cancer cells	Pharmacogenomic data analysis to discover drug–mutation interactions in cancer cells	Liver cancer patients non-responding to sorafenib treatment exhibit higher expression of extracellular sulfatase (SULF2) and LCN2. SULF2 mutation or inhibition enhances sorafenib sensitivity in liver cancer cells.	Diagnostic or therapeutic targeting of SULF2 and/or LCN2 can be a novel precision strategy for sorafenib treatment in HCC	[96]
**Mouse**	Tissue and serum samples from mouse model of HCC	Western blot analysis and immunostaining	LCN2 overexpression in HCC livers in mouse liver, both in transcriptional and protein levels specifically in the tumoral area extracts	LCN2 is a pleiotropic protein, possibly participating in multiple functions in the tumor microenvironment, such as damage response, immunity, and differentiation	[53]
**Mouse**	Implantation of tumors in nude mice	Mice were injected with *Lcn2* overexpressing cell lines (i.e., SK-Hep1 cells)	*Lcn2* expressing cells formed far fewer metastatic nodules in the lungs	*Lcn2* inhibits proliferation, invasion, and metastasis in vitro and in vivo through transcriptional suppression of Twist1 in HCC cells	[7]

## Data Availability

This review includes no original data.

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
