# Peer review of "A Scoping Review on Lipocalin-2 and Its Role in Non-Alcoholic Steatohepatitis and Hepatocellular Carcinoma"

_ijms, 2021, doi:10.3390/ijms22062865_

Round 1

Reviewer 1 Report

The authors presented a review article about Lipocalin-2 in liver disease. The paper is well-written and organized. Although Lipocalin-2 is elevated in various inflammatory diseases and cancers, it is expected to clarify the role of lipocalin-2 in NAFLD and HCC.

Here are the comments.

The authors should describe more about the diagnostic ability of lipocalin-2. AUROC and Se/Sp should also be provided. The authors should compare the diagnostic ability of lipocalin-2 with existing markers (such as FIB-4, elastography).

As the authors mentioned, lipocalin-2 is secreted by various cells; how would we assume elevated lipocalin-2 was originated in liver disease?

From the authors’ point of view, Lipocalin-2 is a pro-inflammatory cytokine. Is there any paper comparing Lipocalin-2 levels in different etiologies? Why is the Lipocalin-2 specific to NAFLD? Hepatitis-C and alcoholic hepatitis also cause liver inflammation and fatty liver disease.

Minor comments.

The name of the last author was typed as Asimakopoulou in the references. Please make sure there is no error.

Author Response

  1. The authors presented a review article about Lipocalin-2 in liver disease. The paper is well-written and organized. Although Lipocalin-2 is elevated in various inflammatory diseases and cancers, it is expected to clarify the role of lipocalin-2 in NAFLD and HCC.

Thank you for your comments. We have now tried to modify/correct all points below to improve our review.

  1. The authors should describe more about the diagnostic ability of lipocalin-2. AUROC and Se/Sp should also be provided. The authors should compare the diagnostic ability of lipocalin-2 with existing markers (such as FIB-4, elastography).

Thank you for your comment. We have now added in the section of ´Lipocalin-2 in NAFLD pathophysiology´ a few lines regarding the points above to complete our description of the diagnostic ability of LCN2 as well the respective reference. All changes are indicated in red color.

  1. As the authors mentioned, lipocalin-2 is secreted by various cells; how would we assume elevated lipocalin-2 was originated in liver disease?

That is indeed a good point. Lipocalin 2 is a pleiotropic marker, however, there is a way to connect its elevated levels to liver disease. Its increase, even though connected to various pathologies, will correlate with other diagnostic markers of a specific disease. To clarify, elevated levels of LCN2 will not directly imply liver disease if not combined with already known markers of liver function as liver enzymes, such as aspartate aminotransferase (AST) and alanine aminotransferase (ALT) that are currently the most sensitive indicators of hepatocyte injury.

  1. From the authors’ point of view, Lipocalin-2 is a pro-inflammatory cytokine. Is there any paper comparing Lipocalin-2 levels in different etiologies? Why is the Lipocalin-2 specific to NAFLD? Hepatitis-C and alcoholic hepatitis also cause liver inflammation and fatty liver disease.

There is unfortunately no paper comparing circulatory levels of LCN2 from different pathological conditions. However, it has been shown in several studies focusing on lung infections, liver damage, hepatectomy, neuroinflammation and blood/brain barrier dysfunction that the main source of serum LCN2 are the hepatocytes supporting a liver-targeted organ axis based pathology [Xu et al., 2015; Borkham-Kamphorst et al., 2013; Mondal et al., 2020]. As you mentioned, in both HCV and alcoholic hepatitis serum and urinary LCN2 has been found indeed to be increased. However, in the case of HCV, urinary LCN2 has been proven to correlate with fibrosis and cirrhotic outcome as well as progress to hepatocellular carcinoma [Kim et al., 2010], while on alcoholic hepatitis LCN2 could indicate progress to fibrosis and portal hypertension [Chen et al., 2020]. Altogether, NAFLD is a pathological condition based on no pathogen or toxic substance overconsumption as alcohol or drugs, criteria that can be excluded directly during patient examination and diagnosis. Under those circumstances LCN2 could indeed serve as a marker for fatty liver disease.

Xu, M. J., Feng, D., Wu, H., Wang, H., Chan, Y., Kolls, J., Borregaard, N., Porse, B., Berger, T., Mak, T. W., Cowland, J. B., Kong, X., & Gao, B. (2015). Liver is the major source of elevated serum lipocalin-2 levels after bacterial infection or partial hepatectomy: a critical role for IL-6/STAT3. Hepatology (Baltimore, Md.), 61(2), 692–702. doi.org/10.1002/hep.27447

Borkham-Kamphorst E, van de Leur E, Zimmermann HW, Karlmark KR, Tihaa L, Haas U, Tacke F, Berger T, Mak TW, Weiskirchen R. Protective effects of lipocalin-2 (LCN2) in acute liver injury suggest a novel function in liver homeostasis. Biochim Biophys Acta 2013;1832(5):660-73. doi: 10.1016/j.bbadis.2013.01.014.

Mondal, A., Bose, D., Saha, P. et al. Lipocalin 2 induces neuroinflammation and blood-brain barrier dysfunction through liver-brain axis in murine model of nonalcoholic steatohepatitis. J Neuroinflammation 2020;17:201. doi.org/10.1186/s12974-020-01876-4

Kim J. W., Lee S. H., Jeong S. H., Kim H., Ahn K. S., Cho J. Y., et al. Increased urinary lipocalin-2 reflects matrix metalloproteinase-9 activity in chronic hepatitis C with hepatic fibrosis. Tohoku J. Exp. Med. 2010;222:319–327. doi.org/10.1620/tjem.222.319

Chen J, Argemi J, Odena G, Xu MJ, Cai Y, Massey V, Parrish A, Vadigepalli R, Altamirano J, Cabezas J, Gines P, Caballeria J, Snider N, Sancho-Bru P, Akira S, Rusyn I, Gao B, Bataller R. Hepatic lipocalin 2 promotes liver fibrosis and portal hypertension. Sci. Rep. 2020;10(1):15558. doi: 10.1038/s41598-020-72172-7.

Minor comments.

  1. The name of the last author was typed as Asimakopoulou in the references. Please make sure there is no error.

Thank you for the observation. This is actually not an error as Mrs Asimakopoulos has recently had her name corrected by official authorities. Therefore, she has since continued publishing her work under the corrected version of her name.

Reviewer 2 Report

I read with interest the review on the role of lipocalin-2 in the setting of NAFLD. Overall, I tink that there are too many and confused informations about lipocalin-2 in different clinical conditions. I think that Table 1 should be eliminated to better focusing on the diagnostic value of lipocalin-2 in NAFLD and NASH-related HCC.

Major comments

  • The english language must be thoroughly revised.  I highly recommend a grammar check. Many sentences lack the subjects and the text comprehension is difficult.
  • Line 39. Remove the semicolon
  • Line 42. LCN2 ... with limited sequence homology... homology with what?
  • Line 50. Quite the opposite, they range... please report the subject of the sentence
  • Line 58. ...could serve as therapeutical...
  • Line 59. Please add the subject at the beginning of the sentence
  • Line 61-62. Plese, clarify
  • Line 64. Consider to replace "to this day" with "to date2
  • Line 67 ...that LCN2 is acting as a ... --> can act
  • Line 77. Consider to replace "Research is scarce..." with "data are scarce..."
  • Line 96. ...toward LCN2 that other ... --> toward LCN2 compared to other ...
  • Line 99. Another is the solute ... Another receptor?
  • Line 114. The expression of lipocalin- 2 in tissue. --> ... tissues
  • Line 130. Non-alcoholic fatty liver disease is considered a clinical condition
  • I think the the paragraph on NAFLD is not necessary and it could be integrated with the next paragraph.
  • Line 41-44. Triglycerides are subjected to lipolysis, not free fatty acids. The results of lipolysis is the hydrolization of triglycerides to free glycerol and FFA. The impaired lipolysis is the consequence of adipose tissue insulin resistence, the main pathogenetic mechanism involved in the onset of NAFL and progression to NASH.
  • Line 147. De novo lipogenesis is increased by 5-fold in NASH patients compared to healthy controls, independently by obesity (Donnelly KL, J Clin Invest 2005).
  • Line 154. It does not exist a condition defined pre-fatty liver. NAFL (or simple steatosis) is the accumulation of triglycerides in the hepatocytes (above 5%) and this condition is reversible through the diet and physical exercise.
  • Figure 1. consider to integrate the spectrum of NAFLD with molecular mechanisms in which is involved lipocalin-2.
  • Line 185-186. ....a base for further pathologies. Which?
  • Line 201. [Tarantino et al.] Add the corresponding reference number.
  • Line 223-224. Add reference
  • Line 228. The study of Milner and colleagues... showed that LCN2 levels correlated with the degree of liver inflammation and the stage of hepatic fibrosis.
  • Throughout the text, use the acronym NASH instead of non-alcoholic steatohepatitis.
  • Line 251. We fed... mice with a methionine and ...
  • I think that Table 1 could be eliminated since the topic of the review is lipocalin.2 in the setting of NAFLD. I suggest to replace this table with another one with all the papers investigating the diagnostic role of lipocalin-2 in NAFLD/NASH as well as in identifying hepatic inflammation of fibrosis.
  • Consider to summerize the conclusions focusing on the most important aspects linking lipocalin-2 with NAFLD and NASH-related HCC.

Author Response

I read with interest the review on the role of lipocalin-2 in the setting of NAFLD. Overall, I think that there are too many and confused informations about lipocalin-2 in different clinical conditions. I think that Table 1 should be eliminated to better focusing on the diagnostic value of lipocalin-2 in NAFLD and NASH-related HCC.

Thank you very much for your comments. We have now tried to improve the review based on all your detailed observations.

Major comments

The english language must be thoroughly revised.  I highly recommend a grammar check. Many sentences lack the subjects and the text comprehension is difficult.

Thank you for your observation. The paper has been now proofread by an English language expert and corrections have been made.

  1. Line 39. Remove the semicolon

We have now removed the semicolon as you suggested.

  1. Line 42. LCN2 ... with limited sequence homology... homology with what?

We here refer to the protein sequence homology between LCN2 and the several members of the lipocalin family.

Line 50. Quite the opposite, they range... please report the subject of the sentence

We have now modified the sentence so that the subject should be clear.

Line 58. ...could serve as therapeutical...

We have now modified the sentence.

Line 59. Please add the subject at the beginning of the sentence

We modified accordingly.

Line 61-62. Plese, clarify

We changed the sentence in a way that we are better understood.

Line 64. Consider to replace "to this day" with "to date2

Replacement adjusted as suggested.

Line 67 ...that LCN2 is acting as a ... --> can act

We have now corrected the sentence.

Line 77. Consider to replace "Research is scarce..." with "data are scarce..."

Thank you, we have now replaced the phrase.

Line 96. ...toward LCN2 that other ... --> toward LCN2 compared to other ...

We have now modified the expression.

Line 99. Another is the solute ... Another receptor?

We have now improved the sentence.

Line 114. The expression of lipocalin- 2 in tissue. --> ... tissues

Correction done.

Line 130. Non-alcoholic fatty liver disease is considered a clinical condition

We have now added the word ´condition´.

I think the paragraph on NAFLD is not necessary and it could be integrated with the next paragraph.

We fully understand your concern. However, we find such a paragraph a necessary introduction to NAFLD before we describe the roles of LCN2 in pathogenesis in more details.

Line 41-44. Triglycerides are subjected to lipolysis, not free fatty acids. The results of lipolysis is the hydrolization of triglycerides to free glycerol and FFA. The impaired lipolysis is the consequence of adipose tissue insulin resistence, the main pathogenetic mechanism involved in the onset of NAFL and progression to NASH.

Thank you very much for the crucial observation. According to your advice, we have now corrected the sentences.

Line 147. De novo lipogenesis is increased by 5-fold in NASH patients compared to healthy controls, independently by obesity (Donnelly KL, J Clin Invest 2005).

We have added the suggested reference.

Line 154. It does not exist a condition defined pre-fatty liver. NAFL (or simple steatosis) is the accumulation of triglycerides in the hepatocytes (above 5%) and this condition is reversible through the diet and physical exercise.

Thank you, we have now modified the sentence.

Figure 1. consider to integrate the spectrum of NAFLD with molecular mechanisms in which is involved lipocalin-2.

Thank you for this comment. However, the following figures 2 and 3 are essentially showing the molecular mechanisms by which LCN2 is involved in NAFLD and HCC. Therefore, we have not added the requested information into figure 1 to avoid repetitions.

Line 185-186. ....a base for further pathologies. Which?

Thank you for your comment. We have now clarified that we are talking for hepatocellular carcinoma.

Line 201. [Tarantino et al.] Add the corresponding reference number.

We have now the respective reference number.

Line 223-224. Add reference

We have now added the missing reference.

Line 228. The study of Milner and colleagues... showed that LCN2 levels correlated with the degree of liver inflammation and the stage of hepatic fibrosis.

We have corrected the sentence as per your suggestion.

Throughout the text, use the acronym NASH instead of non-alcoholic steatohepatitis.

We have now used the acronym throughout the review.

Line 251. We fed... mice with a methionine and ...

We have corrected the sentence.

I think that Table 1 could be eliminated since the topic of the review is lipocalin.2 in the setting of NAFLD. I suggest to replace this table with another one with all the papers investigating the diagnostic role of lipocalin-2 in NAFLD/NASH as well as in identifying hepatic inflammation of fibrosis.

Thank you for your suggestion. The table was implemented to emphasize how complicated it is to define the role of LCN2 in cancer as the review is handling the topic of LCN2 in NAFLD as well as its progress to HCC. That being said, it is easier to explain later on that its role in HCC is no less complicated. Moreover, us and others have already presented tables with the actual up to date roles of LCN2 in NAFLD/NASH including inflammation and fibrosis [Asimakopoulou et al., 2015; Wieser et al., 2016, Asimakopoulou et al., 2016] already cited in the review. Therefore, we prefer to not produce a review with repetitive information. For the reasons above, we would like to keep Table 1.

Consider to summarize the conclusions focusing on the most important aspects linking lipocalin-2 with NAFLD and NASH-related HCC.

Thank you for the comment. We have now modified our conclusion accordingly.

Reviewer 3 Report

In my opinion the manuscript is well structured. The different sections of the text are able to structure a global idea of LCN2 and its role in NAFLD and HCC. The review figures are clear and summarize the information presented in the text. In the same way, the tables summarize the clinical findings associated with LCN2 in cancer models, tissues and cell lines (also the clinical findings associated with LCN2 in hepatocellular carcinoma).

I have not detected any excess of citations of the authors in the manuscript, knowing that the authors work in the research field.

In figure number 2 (in my revised version) in the effect box appears the term: de novo lipogenesis, with a line ???. Please check this format. 

Author Response

  1. In my opinion the manuscript is well structured. The different sections of the text are able to structure a global idea of LCN2 and its role in NAFLD and HCC. The review figures are clear and summarize the information presented in the text. In the same way, the tables summarize the clinical findings associated with LCN2 in cancer models, tissues and cell lines (also the clinical findings associated with LCN2 in hepatocellular carcinoma).

I have not detected any excess of citations of the authors in the manuscript, knowing that the authors work in the research field.

Thank you very much for kind words.

  1. In figure number 2 (in my revised version) in the effect box appears the term: de novo lipogenesis, with a line ???. Please check this format. 

You are correct. In Figure 2 the line is intentional there to indicate that LCN2 elevation or overexpression causes the cessation of de novo lipogenesis, while it enhances several other processes mentioned in the figure. To avoid confusion to the readers, we have replaced the strikethrough line with a pointing down arrow indicating what we mean.

Round 2

Reviewer 1 Report

The authors have modified the manuscript as requested. I do not have further comments.

Reviewer 2 Report

I agree with all the modifications made by the authors and I have no further comments.